# Long-Term Thermal Comfort Monitoring via Wearable Sensing Techniques: Correlation between Environmental Metrics and Subjective Perception

**DOI:** 10.3390/s23020576

**Published:** 2023-01-04

**Authors:** Veronica Martins Gnecco, Ilaria Pigliautile, Anna Laura Pisello

**Affiliations:** 1CIRIAF—Interuniversity Research Center on Pollution and Environment Mauro Felli, University of Perugia, 06125 Perugia, Italy; 2Engineering Department, University of Perugia, 06125 Perugia, Italy

**Keywords:** thermal perception, clustering process, wearable sensing, personal exposure, energy efficiency, thermal comfort, multidomain comfort, indoor environmental quality, IEQ

## Abstract

The improvement of comfort monitoring resources is pivotal for a better understanding of personal perception in indoor and outdoor environments and thus developing personalized comfort models maximizing occupants’ well-being while minimizing energy consumption. Different daily routines and their relation to the thermal sensation remain a challenge in long-term monitoring campaigns. This paper presents a new methodology to investigate the correlation between individuals’ daily Thermal Sensation Vote (TSV) and environmental exposure. Participants engaged in the long-term campaign were instructed to answer a daily survey about thermal comfort perception and wore a device continuously monitoring temperature and relative humidity in their surroundings. Normalized daily profiles of monitored variables and calculated heat index were clustered to identify common exposure profiles for each participant. The correlation between each cluster and expressed TSV was evaluated through the Kendall tau-b test. Most of the significant correlations were related to the heat index profiles, i.e., 49% of cases, suggesting that a more detailed description of physical boundaries better approximates expressed comfort. This research represents the first step towards personalized comfort models accounting for individual long-term environmental exposure. A longer campaign involving more participants should be organized in future studies, involving also physiological variables for energy-saving purposes.

## 1. Introduction

During people’s daily activities, different environmental stimuli impact their well-being and comfort perception and, consequently, their work productivity [1], health [2], and happiness [3]. Measuring and understanding the triggers that lead to human sensations can help to calibrate and configure ideal conditions [4]. Even if people spent almost 90% of their lifetime in indoor environments [5], outdoor stimuli also influence overall human wellness. In this case, people are exposed to conditions that are more difficult to be modified and controlled, when compared to buildings equipped with HVAC systems, lighting appliances, and performing construction materials. Moreover, the human perception of comfort also affects their interaction with the environment and buildings’ operation.

Furthermore, the definition of thermal comfort provided by ISO 7730 [6], which can be adapted to the other perception domains, stated that it is the “condition of mind which express satisfaction with the thermal environment”. Comfort studies demand the quantification of a qualitative indicator that is the satisfaction level of the occupants, to possibly drive improvement of indoor, or even outdoor, conditions. In this context, comfort models assist international standards to quantify human comfort perception, in different dimensions. The Predicted Mean Vote (PMV) and the Percentage of People Dissatisfied (PPD) indices, developed by Fanger in the 1960s, are widely adopted to access indoor thermal comfort in research, considering the simple approach and satisfactory results [7]. The indices construction is based on laboratory tests, which combine the human physiological responses to environmental stimuli and the theory of heat balance for thermoregulation. The PMV model adopted a seven-point scale to predict the mean value of human subjective thermal sensation votes, in indoor comfort studies. This scale represents the preferences of a group of people occupying the same building, going from −3 (cold) to +3 (hot), where 0 corresponds to the neutral thermal sensation [8]. However, it should be addressed that to predict theoretically the thermal sensation of occupants is not an easy task, considering that steady-state conditions are rarely found in real environments and other external stimuli can also influence the thermal perception of users, such as activity level, occupancy, and other multi-domain interactions [7].

On the other hand, the adaptative thermal comfort model tried to explain the divergences between the classic PMV-PPD approaches and the actual thermal sensation in naturally ventilated buildings, through a simple regression model that relates the indoor and outdoor temperatures. International Standards used the adaptive approach as a conceptual foundation and guidance, such as ASHRAE 55 [9], ISO 7730 [6], EN 16798 [10], and CIBSE Guide A [11]. ASHRAE 55 Standard, for example, recommends that indoor temperatures should lead the PMV index to a value between −1 and +1, in order to reach only 25% of the occupants dissatisfied. The ISO 7730 describes an equation to determine the PMV value, based on environmental variables and physiological characteristics. Nevertheless, the PMV scale has been questioned and adapted several times, considering that non-neutral thermal preferences are common among occupants [12].

Martins, Soerbato, and Williamson [12] investigated different scales used to translate thermal preference or sensation, across 37 studies. Depending on the study’s approach, some scales were converted to a lower number of points. Jazizadeh et al. [13] used a 5-point scale to determine occupants’ personalized comfort profiles and optimize the control of the Heating, Ventilation, and Air-Conditioning (HVAC). The new scale was adapted from the 7-point scale, considering that the intermediate points (−1, 0, and +1) are all considered satisfactory by the standards. The scale adoption and configuration were investigated by Schweiker et al. [14], addressing the relationship between temperature and subjective thermal sensation. The authors concluded that the type of scale used to collect the occupants’ perceptions impacts the results and that the scale range of comfortable differs from person to person. Therefore, the association of these subjective interpretations with measurable indices, such as environmental [15] and physiological [16] information, can bring a more integrated and complete interpretation of human comfort perception. Some studies have already tried to overcome the weaknesses of thermal comfort models, since they were developed for a group of people, not for individual preferences.

This association can identify patterns in the subjects’ perception and consequently drive the ambient environmental control toward more comfortable conditions. In this case, personal comfort models, based on people’s perception of comfort and on the environmental conditions which shape satisfaction, can be drawn and help the creation of a comfortable environment with the reduction of energy use [17]. The automation of HVAC systems reduces the issues caused by users’ low awareness of their own thermal sensation, avoiding overheated or overcooled buildings [18]. Yang et al. [19], developed a new technology to measure the skin temperature of occupants without direct contact. In this study, HVAC systems could have their regulation totally demand-oriented, based on the previous calibration of the systems through monitoring of physiological and environmental signals and association with subjective reports of comfort. A similar approach was applied by Deng [17] using wristbands signals and by Cottafava et al. [20], in a crowdsensing and feedback model developed for an office building in the context of the ComfortSense project.

Different sensors and devices are adopted to verify the environmental and physiological dimensions of indoor and outdoor spaces and human responses, respectively. In this view, wearable sensors are particularly suitable for catching the personal human experience of comfort.

As described by Salamone et al. [15], wearable sensors can monitor different domains of comfort, namely thermal, visual, acoustic, and air quality, and catch indoor environmental quality. Moreover, the combination of the different environmental factors helps for the deep comprehension of the external triggers of human sensation. The data collected from environmental and physiological wearable sensors can support machine learning models to predict the thermal comfort perception of occupants [21,22,23]. In this case, data-driven models could optimize the number of experiments done for a certain situation. In any case, the use of wearable sensors to monitor the subject’s daily activities for a long-time assist in understanding their responses to different outdoor stimuli. This database can be updated according to new information it receives from users to the future improvement of indoor conditions and perceived comfort.

Connecting the different spheres of comfort is the best option for detecting real human experience. Nazarian et al. [24] proposed a wrist-mounted wearable sensor to measure microclimate parameters, and physiological, and subjective answers from the subjects, to access human thermal comfort and heat stress. The tool proved to be satisfactory to predict body core temperature and the overall sensation of participants. Mansi et al. [25] investigated the encephalographic (EEG) signals using wearable devices, during an experimental campaign in a test room. The subjects were exposed to different thermal conditions and answered different surveys relating to their comfort perception. The environmental characteristics were also monitored. Even presenting reliable results, the wearable device required a processing step.

Outdoor environments were also studied in the last few years, considering the fast growth of urban areas. Pioppi, Pisello, and Ramamurthy [26] used a wearable apparatus to collect key environmental parameters for pedestrian comfort in urban parks. The sensors indicated that urban parks could help to reduce the effects of heat islands in cities. Biosensors are another type of tool that could detect small toxic molecules, pollutants, and life-threatening agents present in inhabited environments [27]. Essentially, portable and wearable devices have lower performance than medical [28] and fixed devices [15], but recent advancements are being made to increase their accuracy, through improvements in sensing technology and data processing [16].

However, there is still a lack of long-term studies to investigate the human comfort experience. Liu et al. [29] carried out an experiment monitoring fourteen subjects for at least fourteen days, using different wearable devices: iButton hygrochron to monitor the external air temperature and relative humidity; iButton hygrochron to the skin temperature at the ankle and wrist; a chest strap to the heart rate and a smartwatch to wrist accelerometry. The authors were able to use the data collected, together with comfort survey answers, to establish personal comfort models more reliable than the traditional PMV and adaptive comfort models. Ji et al. [30] also conducted a long-term study using wearable devices, for less than two months, to understand the influence of the immediate thermal history of the subjects on the current thermal history. The participants were asked to wear a portable probe outside their backpacks to register temperature and humidity data and to give feedback on thermal sensation whenever they experience some abrupt change of condition or position. The temperature acceptability range was wider than expected and the cold temperatures had a higher influence on the thermal perception. 

Research that performs data acquisition for a longer time should have well-established procedures for data gathering and storage, considering the high amount of information collected. Wearable devices are practical and non-intrusive solutions, usually available commercially [31]. The connectivity of wearable sensors with the virtual world is promoted for the Internet of Things (IoT), improving interoperability, communication, and response dynamism, to create a unified environment [32]. IoT solutions are growing alternatives to traditional methods of monitoring and calculation of environmental data and can assist in the promotion of a more human-centric approach to build environment research [33].

Salamone et al. [34] performed data collection using wearable sensors, machine learning techniques, and IoT to assess human thermal perception and improve the environmental condition of the surroundings accordingly. The use of IoT solutions allowed the researchers to build personal thermal comfort models considering the characteristics of the indoor environment monitored. Future implementations foresee the use of real-time data to adjust indoor conditions according to the users’ thermal preferences. The campaign carried out by Liu et al. [28], for example, allowed researchers to cloud computing data and use this information for real-time prediction of occupants’ thermal preferences.

Using wearable sensors, devices are more and more accessible, it is possible to access high-granularity data from occupants and understand the triggers of discomfort perception. The inclusion of different parameters and real-time information in comfort models enables the development of realistic predictions, controlled by IoT appliances in a more efficient way [31]. In the current scenario of thermal comfort research, the detection of thermal comfort parameters is fundamentally based on camera technologies, wearable devices, and fixed sensor systems. 

However, a point still underexplored is the acceptability of occupants to the wearable technology and the ethical concerns for this data employment for research. Usually, physiological data collected from users can raise more privacy concerns than environmental ones, and their acquisition should be studied and agreed upon among all participants [35]. The same concern emerges when it comes to IoT devices connected with buildings, highly dependent on individual information to develop personal comfort models. Aside from the ergonomic design and their integration into the occupants’ daily activities [15], users should understand the practical benefits of the data collection and how it can improve their productivity, happiness, and health [36]. Nevertheless, the study of different factors that lead to discomfort, relating them with human perception in a long-term approach, could bring insightful conclusions about the actual environment that people are daily exposed to.

Considering the previously presented knowledge gaps, this study aims to investigate the occurrence of a correlation between mean daily Thermal Sensation Vote (TSV) and different normalized metrics of air temperature, relative humidity, and heat index, i.e., mean minimum and maximum values. All the data were collected during long-term monitoring, through a wearable device, expecting to understand the relationship between the human’s subjective sensation of thermal comfort and the variation of monitored environmental values.

Therefore, a potential contribution can be identified by relating conditioned controlled environments, in terms of temperature and relative humidity, and the comfort sensation of the subject experiencing these environments: do only the absolute values of temperature and relative humidity impact the thermal sensation of occupants, or their fluctuation have a stronger effect? The occupants can, at the same time, adapt to the current environmental conditions and be more influenced by greater modifications in a shorter time. Understanding which factors are more influential and whether the difference of its amplitude during the day determines an important factor for the long-term comfort perception of occupants, can represent a first step to the implementation of personal comfort models and guide the use of active strategies for improving thermal comfort while reducing energy use.

Considering the proposed objectives and contributions provided to the human-centric research, this paper is organized as follows: Section 2 describes the method adopted to collect, analyze and process the data; Section 3 presents the results from the long-term monitoring; Section 4 discusses the produced outcomes and Section 5 resumes the study and organizes final comments and future perspectives.

## 2. Materials and Methods

The proposed method aims to understand the relationship between human thermal sensation and different environmental indices. The data gathering was enabled using wearable sensors, which collected air temperature (Ta) and relative humidity (RH) information continuously during the campaign period. The adopted procedure was divided into three steps: (1) data collection characterization; (2) data analysis and clustering process and (3) statistical evaluation, which will be detailed in the following sections. Figure 1 describes graphically the method.

### 2.1. Data Collection Characterization

During the data collection, six people wore one iButton Hygrochron (Figure 1) for Ta and RH monitoring. The participants were instructed to leave the sensors visible to the outdoor environment and attached to accessories of daily use, such as key chains, backpacks, or even coats. It was also recommended that the sensors should not be exposed to direct sunlight, to avoid overheating or being immersed in waterbodies, considering that they are water-resistant, but not waterproof. Alarms were set to send reminders to the participants, according to the usual hour that they leave home. The participants’ training ensures the proper usage of the sensors during the experiment, avoiding data loss or miscollecting. Unlike most human comfort experiments, in this case, the subjects are daily monitored during the whole experimental period and are responsible for the correct use of the probes, without the constant assistance of the researchers.

The iButton Hygrochron is a portable sensor available commercially, with 8 KB of data-log memory and software for setup and data retrieval. The timestep and initial time of recording are configurable through this user interface, as well as the data saving. The Ta and RH parameters collected by the probe have an accuracy of ±0.5 °C and ±5%, respectively. The sensor tolerates temperatures from −20 °C to 85 °C [29]. The information was registered assuming a timestep of five minutes. Before starting the data collection, the participants agreed to cooperate with the activity by signing a consent form, declaring to understand the procedure and that their personal information will not be disclosed during this research. All the subjects are volunteers in good health condition, i.e., not affected by any circumstances that could affect the environmental perception and the survey results. The monitoring activity occurred in indoor and outdoor environments, for 24 h, 7 days a week, for at least one month in different seasons of the year.

The participants were also asked to answer a survey relating their perception of different comfort dimensions, i.e., thermal, acoustic, visual, and air quality, three times a day: right before waking up, between 12 PM and 14 PM, and before going to sleep. A question concerning the stress level was also asked, considering the same scale used for the comfort sensation. In this study, only the thermal comfort sensation was considered. A five-point scale was adopted, presented in Table 1.

Table 2 presents the start and end date of data collection for each subject. However, it is important to point out that some days between these dates were not registered due to data rollover issues. The last column of Table 2 gives the percentage of the received answers per the total expected answers in the daily surveys. Summer and spring were related to the “hot seasons”; autumn and winter to the “cold seasons”.

Due to the lack of sufficient survey answers, the data from participant VI, described in Table 1, had to be discarded from the analysis. Thus, only five people remained in the final sample. 

Aside from the daily survey and the monitoring devices, the study had no further impact on their daily routine. The participants involved were women, between 25 and 35 years, working in office jobs and with quite similar routines. The working hours were considered from 9 AM to 7 PM, based on the related daily schedule of the participants. The metabolic rate for each person was calculated using Annex C from ISO 8996:2021 [37], varying from 450.88 W to 560.98 W, with a standard deviation of ±57.47. The mean heart rate, required for the calculation, was obtained from the Samsung Galaxy smartwatch provided to the subject for the whole experimental campaign. According to previous research, the device has an accuracy of around +/− 4 bpm [38,39]. Table 2 shows the main information for each subject at the time of the data collection.

The study was carried out in Perugia, Italy. According to the Köppen-Geiger criteria, Perugia has an overall mild climate, with no dry season and constantly moist, classified as a humid subtropical climate (Cfa) [21].

### 2.2. Data Analysis and Clustering Process

Due to the large amount of collected data, Python routines were used to optimize the data processing step. First, a cleansing process was carried out, to remove outliers from the database, for both *Ta* and *RH* [40]. Equations (1) and (2) were adopted to establish the lower and upper limits, respectively.
 Lower limit = 1º Quartile − 1.5 × (3º Quartile − 1º Quartile)(1)
 Upper limit = 3º Quartile + 1.5 × (3º Quartile − 1º Quartile)(2)

A normalization process was performed to determine the different profiles of hourly *Ta* and relative humidity amplitude during the day, considering three different indices: mean, minimum and maximum values. The normalization allows recognizing only the metrics amplitude, looking in detail at the effect of their variation, not the absolute values. Additionally, by calculating the normalized values, it is possible to compare data from different periods and seasons of the year. The 24-h profiles were calculated, and the correspondent minimum daily *Ta* was subtracted from each hourly value, for the different *Ta* and *RH* metrics. For example, considering the maximum hourly *Ta*, the value subtract was the minimum maximum *Ta* for each day. The reasoning was the same adopted for the mean and minimum values. Equation (3) represents the calculation performed for the *Ta* and Equation (4) for relative humidity.
(3)$$Ta_{n,h,\;\left(mean,min,max\right)}=Ta_{\left(mean,min,\;max\right),h}-Ta_{min,d,\left(mean,min,max\right)}$$

Where *Tan,h* is the hourly normalized mean, minimum or maximum temperature (°C), *Tamean,h* is the hourly mean, minimum or maximum temperature (°C) and *Tamin,d* is the minimum temperature for the corresponding day for the mean, minimum or maximum profiles (°C).
(4)$$RH_{n,h,\;\left(mean,min,max\right)}=RH_{\left(mean,min,\;max\right),h}-RH_{min,d,\left(mean,min,max\right)}$$
where *RHn,h* is the hourly normalized mean, minimum or maximum relative humidity (%), *Rhmean*, *h* is the hourly mean, minimum or maximum relative humidity (°C), and *Rhmin,d* is the minimum relative humidity for the corresponding day for the mean, minimum or maximum profiles (°C).

Calculations were also performed to obtain the heat index metric (*HI*) [41], an index that combines temperature and relative humidity values to determine apparent temperature (Equation (5)). The values for *Ta* and *RH* collected by the data acquisition were used to calculate the *HI*, which was later normalized as the other two metrics.
(5)$$HI=-8.784695+1.61139411\times T+2.338549\times RH-0.14611605\times T\times RH-1.2308094\times 10^{-2}\times T^{2}-1.6424828\times 10^{-2}\times RH^{2}+2.211732\times 10^{-3}\times T^{2}\times RH+7.2546\times 10^{-4}\times T\times RH^{2}-3.582\times 10^{-6}\times T^{2}\times RH^{2}$$

The days without any answers to the comfort perception survey were excluded from the database, considering that in the next step, the statistical evaluation, both normalized metrics (*Ta* and *RH*) and TSV values are necessary to perform the correlation.

According to the remaining hourly normalized 24 h temperature and relative humidity profiles for each day, the k-means cluster method [42], with 100,000 iterations, was adopted to classify the different daily profiles (lower to a higher variation of hourly temperatures, *RH* and *HI*). To determine the optimum number of clusters for each case, two criteria were considered: (1) the number of elements per cluster and (2) the within-cluster average distance (the within-cluster sum of squares, *WSS*). The values for these two criteria were calculated for a predefined number of clusters k from two to eight in each case (Figure 2). The optimum number of clusters was decided considering the option with at least 10 elements per cluster and a low average distance from the cluster elements to their centroid (average within-cluster sum of squares), according to Equation (6).
(6)$$WSS_{m}=\frac{\sum_{i=1}^{n}{\left(x-x_{i}\right)}^{2}}{n}$$
where *WSSm* is the average within-cluster sum of squares, *n* is the number of observations, *xi* is the observation value and *x* is the centroid position for the corresponding day and hour.

The minimum number of elements per cluster follows the sample size calculation for the Kendall-tau b test, explained by Bonett and Wright [35,43], with a confidence level of 0.90, 0.30 Fisher confidence interval, and 0.01 confidence interval.

This procedure was adopted for each subject separately. The latter was considered due to the homogeneity of the sample, which may provide some pattern behavior for the group. Considering the five participants, three different variables, i.e., *Ta*, *RH*, *HI*, and the three different metrics, i.e., mean, minimum, and maximum, 45 clustering procedures were performed after the determination of the optimum cluster number in each case.

### 2.3. Statistical Evaluation

Finally, considering the optimum number of clusters, a Kendall tau-b test was performed through Python routines, to verify the correlation between the daily normalized profiles (mean, minimum and maximum values for temperature and relative humidity data) and the mean daily TSV. The Kendall tau-b is a correlation test, which measures the strength and direction of association between two variables organized on an ordinal scale. The Kendall tau-b test is an alternative for both Pearson’s and Spearman’s tests, considering its non-parametric nature and that it allows tests with smaller sample sizes and many tied ranks, respectively. The Kendall tau correlation coefficient varies from −1 to 1, where −1 indicates a perfect negative association between the two variables, 1 is a perfect positive association between the two variables, and 0 is independence between the variables. However, values higher than |0.35| and between |0.21| and |0.35| can be already considered with a strong and moderate association between the analyzed variables, respectively.

To understand the behavior of each cluster, the daily mean values of the normalized Ta, RH, and HI data were also calculated. Then, lower or higher clusters mean amplitudes can be verified when a correlation between the metrics and the TSV is identified, and a possible pattern is established. The procedure was repeated for each cluster and subject.

## 3. Results

### 3.1. Overview and Clustering Process

Data collection is a fundamental step to the success of human-centric research. The daily participants’ survey answers, for example, allowed the correlation between this subjective metric and environmental parameters, for the statistical test. However, only 70% of the total expected answers were completed by the participants of this study. Figure 3 shows the overall mean daily thermal sensation by subject. All the subjects present median thermal sensation values within the thermal neutrality range established in ASHRAE 55 [36], i.e., values between −0.5 to +0.5. The extreme values, with +2 representing hot and −2 cold, were chosen by only 2.03% considering all the subjects, indicating favorable thermal conditions or broader thermal acceptability from the participants. In general, the range of thermal sensation values does not vary significantly between the subjects, supporting the premise of homogeneity between the group: minimum and maximum values were always between +1 and −1. Calculating the thermal deviation from the center (neutral sensation), the values found were similar between the subjects, respectively 0.090, 0.081, 0.082, 0.069, and 0.090 for subjects I, II, III, IV, and V. Subject IV presented the lowest deviation and subjects I and V the highest ones.

The number of samples for each subject also directly influences the number of clusters and, obviously, the number of elements by cluster. Based on the criteria presented in the Section 2, the number of clusters for each subject was calculated for temperature, relative humidity, and heat index, respectively shown in Figure 4, Figure 5 and Figure 6.

In two cases for the temperature clustering, only one cluster was selected, i.e., when having two or more clusters the number of elements by cluster was lower than 10. This could be observed using the minimal and mean normalized temperature values for the II and V subjects. The possible explanation, analyzing the data from each one of the subjects and based on the k-means clustering method, it is the similarity between the daily profiles obtained in these two cases, not reaching more than 10 elements by a cluster when the process is performed. Having just one cluster, the average within-cluster sum of squares for these two cases is slightly higher when compared with the others, presented in Table 3.

Winter and summer seasons were not differentiated, as the analysis concerned the normalized temperature and relative humidity values, not the absolute ones. The values of within-cluster average distance were similar for all the participants. It is possible to notice that the subjects with more valid data, i.e., III and IV, also have usually a higher number of clusters and lower WSS. 

### 3.2. Clustered Daily Environmental Profiles

In most cases, the clustering process, preceded by the temperature and relative humidity normalization, demonstrated one cluster by metric with more constant temperatures and relative humidity during the day. For the other cases, amplitude turning points, for both temperature and relative humidity values, were observed mostly at 9 AM and 8 PM, i.e., respectively the beginning of working and resting hours according to the subjects’ routine. The range of higher and lower amplitude values depends mainly on the access of the subject to conditioning systems at home or in the office.

For subject I, some differences could be observed between the different metrics: employing the temperature mean values, cluster 0 presented more constant values with low-temperature variation, i.e., mean temperature of 1.20 °C for both working and resting hours, and cluster 1 had a higher variation during the working than resting hours, respectively 3.02 °C and 2.15 °C mean temperature values.

The minimum values of temperature provided two clusters, one higher variation during the resting hours and another one during the working hours, having their peak mean values of temperature variation at 3.68 °C at 2 AM and 3.80 °C at 5 PM, respectively. For the maximum values of temperature, the higher variation was during the working hours for the two clusters, with mean values 0.91 °C and 1.05 °C higher than during the resting hours for clusters 0 and 1, respectively. In this case, it can be noticed that, when the mean values of temperature were used, the temperature amplitude was more constant during the day, probably better expressing the daily overall subject’s temperature perception. 

For RH, one cluster of each metric showed higher variation between 8 PM and 9 AM, with a mean value of 3.58 °C among the metrics during the working period, and another with constant temperatures during the whole day, with a standard deviation of 0.39 °C between the metrics using mean, minimum and maximum temperatures, characterizing two distinct expositions for the subject I. For the normalized HI for the subject, I presented lower and more constant values for the daily profiles, with a standard deviation of xxx for the mean values among clusters. Figure 7 shows the mean daily profiles for temperature (a), relative humidity (b), and heat index (c) for subject I.

For subjects II and V, the trends for temperature values were similar, having for all metrics and clusters higher amplitude during working hours than resting ones, with mean values of 2.41 °C and 1.67 °C, respectively. This behavior indicates a routine during the different seasons of data collection and probably access to conditioning systems during resting time, at home. Subject III presented a similar scenario for at least one cluster by metric, and another with lower and constant temperature amplitude. Subject IV, the one which held the higher volume of data, also presented a higher number of clusters and scenarios by metric, with a higher standard deviation of mean amplitude values when using the minimum metric, compared to mean and maximum ones, respectively 2.49 °C, 1.81 °C, and 1.37 °C. Each metric produced one cluster with lower and constant amplitudes. Concerning the other clusters, most parts of the temperature variation occurred between 9 AM and 8 PM.

Concerning relative humidity, subjects II, III, IV, and V presented one scenario by metric with lower and constant amplitudes and the others with lower amplitude values between 9 AM and 8 PM and higher between 8 PM and 9 AM, for all the analyzed metrics.

Using the Heat Index, the normalized values remained more constant by cluster, having a less clear pattern of variation between working and resting hours. Knowing that this apparent temperature could explain why the participants demonstrated a lower variation from the neutral condition, described in Figure 3, as the index is used to determine how the temperature and humidity together impact human perception. Figure 8 shows the daily profiles for temperature (a), relative humidity (b), and heat index (c) for subjects II, III, IV, and V.

Constant and lower temperature amplitudes can indicate two different exposition situations: either the subject remained all day in the same environment, probably with access to a conditioning system, or the sensor was not brought together with them during its daily activities. Considering that the data collection was carried out during the COVID-19 pandemic, i.e., summer 2020—winter 2021, and that the subjects mainly perform office activities, there is a high possibility of remote work at the time of data acquisition. The data collected can also assist to understand the subjects’ daily routine and the time spent inside or outside the office.

### 3.3. Statistical Analysis for Correlation Calculation

The results obtained for the Kendall tau-b correlation between normalized temperature, relative humidity, and heat index values are summarized in Table 4, Table 5 and Table 6, respectively. Kendall tau-b correlation coefficient, *p*-value, and standard deviation by cluster, concern only the days with both temperature and TSV values. The percentage of received answers was shown previously in Table 2.

For the temperature, the test showed a strong correlation between the variables in five cases: subject I, minimum values, cluster 0; subject III, mean values, cluster 2; subject III, minimum values, cluster 1; subject IV, mean values, cluster 0; subject V, maximum values, cluster 0. These cases were highlighted in green color in Table 5, with mean values for temperature amplitude respectively 1.845, 4.426, 5.264, 1.070, and 1.671. Values are statistically significant when *p*-values are lower than 0.05, i.e., minimum values for cluster 0 of subject I, minimum values for cluster 1 of subject III, mean values for cluster 0 of subject IV, and both mean and maximum values for cluster 0 of subject V.

Moderate correlation arose seven times, highlighted in yellow color in Table 4: subject II, mean values, cluster 2; subject III, mean values, cluster 0; subject III, maximum values, cluster 1; subject IV, mean values, cluster 2; subject IV, maximum values, cluster 0; subject IV, maximum values, cluster 3; subject V, mean values, cluster 0. The mean values for temperature amplitude were respectively 3.322, 1.060, 3.962, 4.982, 1.152, 4.725, and 1.878. These results might show a tendency of correlation between lower temperature amplitudes and TSV. In any case, even with similar sample sizes and temperature standard deviation values, not all cases present the same results. 

Considering the results for relative humidity, three cases presented a strong correlation between the variable and the TSVs, highlighted in Table 5 in green color: subject I, maximum values, cluster 1; subject IV, maximum values, cluster 1; subject V, maximum values, cluster 1. The mean values for relative humidity amplitude were respectively 12.329, 10.007, and 14.072. Moderate correlation, highlighted in yellow color in Table 5, was observed only for subjects I and II, in five cases: mean values, cluster 1, and minimum values cluster 1 for the former and clusters 0, 1, and 2 for the latter. The mean values for relative humidity amplitude in these cases were respectively 10.259, 13.129, 8.509, 11.958, and 17.476. Using maximum values for cluster 1 of subjects IV and V, statistically representative values were obtained for the relative humidity.

Thus, there is little evidence that the daily temperature or relative humidity variation and the daily mean TSV have any correlation, either positive or negative. However, the test must be replicated with higher sample sizes, increasing the data collection period, and keeping track of the participants’ survey answers, to reduce the days without answers.

The Heat Index (Table 6) presented at least one case with a strong correlation for all subjects, except subject IV, without a clear pattern between clusters with higher or lower HI mean amplitude. However, the thermal perception of subjects I, II, III, and V was demonstrated to be closely related to the apparent temperature calculated by the HI. Subject IV, also the one with the lowest deviation from the neutral condition, did not indicate any correlation between the HI and the daily mean TSV values, which may be related to a low sensibility to environmental conditions. The heat index values presented the following statistically significant values: subject I, mean values, cluster 1, and subject II, maximum values, cluster 0.

This section may be divided into subheadings. It should provide a concise and precise description of the experimental results, their interpretation, as well as the experimental conclusions that can be drawn.

## 4. Discussion

Considering the data collected, it is possible to notice a stronger tendency of correlation between the heat index and the thermal sensation vote of the subjects, without a clear pattern of correlation among the different used metrics or the clusters. Subject IV, unlike all the other four participants, did not present any correlation in this case, probably due to a lower sensibility to environmental changes or favorable thermal conditions. Considering that the heat index can represent the apparent temperature and, consequently, better describe the human perception of the environment nearby. Therefore, no clear tendency of correlation between the temperature or relative humidity amplitude and thermal sensation values could be established for the subjects, through the long-term monitoring campaign.

The simple method allowed the inclusion of wearable sensors in the participants’ routine with minimal adjustments and almost any complaints on their side. However, some factors that could have influenced the results should be addressed, such as the possibility of inaccurate data collection, in the cases where the subjects left the sensors at home, registering temperatures with lower amplitude, even if the actual perceive temperature is different. In this research, the participants were instructed to answer the questionnaire just by having it with them, in order to avoid such problems. Subjects should be correctly instructed and regularly remembered about simple and strategic ways to always carry the sensors right positioned with them. Furthermore, the consequences for the research results, caused by poor data acquisition, should be explained to engage participants in self-controlling the sensing system. In this type of data acquisition, in which the researchers are not present the whole time with the subjects, the subjects should be motivated and followed closely as it is possible.

Still, the proposal can be an initial approach for the use of wearable sensors for human-centric data acquisition in a long-term application and in the implementation of personal comfort models through machine learning techniques. The complex nature of people’s comfort perception should be studied considering the whole exposure context, not only under controlled boundaries in conditioned rooms, but also under a climate-dependent outdoor environment. The addition of other variables and dimensions of the human experience, such as physiological measurements with wearable smart watches, can improve even more the study of the long-term exposition influence on people’s thermal perception and preferences.

Additionally, solutions to remind the participants to answer the survey, such as notifications implemented with the smartwatches as in the CoolBit Project [24], should be adopted to avoid data missing. Having a broader sample can help to understand the subjects’ responses to different environmental conditions and more consistent data. For outdoor environments, alerts implemented in wearable sensors can warn people about adverse conditions.

The clustering process helps to understand better subjects’ routines, acceptability to different environmental conditions and possible alterations, and access to cooling or heating systems. Long-term data monitoring with portable devices can also help to develop personal and personalized thermal comfort models, as established by Liu et al. [28], which coupled with HVAC systems, collaborate to a reduction of energy use, while maintaining high comfort levels. The authors used the collected environmental parameters, correlated to subjective and physiological data to train machine learning models and predict subjects’ thermal preferences. The lower sensor intrusiveness can improve the data collected volume to build more accurate models. Still, low prediction accuracy regions could be found due to the multi-factor influence on subjects’ perception. This issue can be addressed by the inclusion of different sensors and by increasing the sample or the time of data acquisition. Additionally, wearable sensors can assist in the implementation of strategies to receive instantaneous feedback from the users and facilitate model adjustments.

## 5. Conclusions

The use of wearable sensors to describe people’s perceived environmental conditions can help to fully understand the triggers of human comfort. This study proposed a long-term monitoring campaign, implementing a human-centric perspective. Five participants were asked to wear an individual sensor for at least one month in different seasons of the year. The sensors collected environmental information in indoor and outdoor spaces, i.e., temperature and relative humidity. The human perspective was accessed through daily surveys, concerning thermal comfort sensation. A clustering process was performed to obtain 24-h temperature, relative humidity, and heat index amplitude profiles for each participant. In each case, three different metrics were used to obtain hourly mean values, the mean, minimum and maximum values, all of them normalized using the daily lower values found for the series. Finally, the Kendall tau-b tests were carried out between the daily normalized metrics and the TSV, for correlation verification by cluster.

Subject I presented the lowest amplitudes for temperature and HI. The other subjects, i.e., I, II, III, IV, and V, presented more similar behavior, with higher variations of temperature and relative humidity between the working hours. Concerning the HI, the values were more regular during the daily period. The clustering process showed in all cases the tendency to have at least one scenario by metric with lower variation, probably related to longer stays in indoor conditioned spaces. The HI metrics presented more cases with a strong correlation with the TSV for all subjects, except subject IV. Subject IV demonstrated the lowest deviation from the neutral conditions in the TSV, possibly caused by high acceptability or low perception of environmental changes. These results are probably related to a better description of the environmental conditions by the index, which determines the apparent temperature relating to temperature and relative humidity data.

The procedure can access even more effective outcomes in campaigns with a higher amount of data, for longer periods of monitoring, and including more participants. Future studies concerning this matter should foresee the requested sample size to obtain more conclusive results. Given the human-centric approach employed in this campaign, strategies to keep the participants engaged in it, i.e., wearing the sensors and answering the survey, should be implemented, guaranteeing serious and active participation of the subjects. Participants must be instructed about the concept of the campaign and how the data is collected, to avoid jeopardizing the results. The data obtained should also be implemented in machine learning techniques to find preference patterns and guide HVAC systems management. Other types of sensors can be adopted, measuring more parameters of the environment, such as biosensors.

Additionally, a more heterogeneous sample can be explored in future studies, seeking general and not localized patterns. People with different routines, metabolic rates, and ages may provide more meaningful information and, consequently, different results. During this campaign, other variables were monitored and can possibly be included in the next applications, such as heart rate data and survey answers for visual, air quality, and acoustic comfort domains, stress level, and sleep quality. Other parameters can refine the sample characterization and allow another type of analysis, such as the introduction of refined personalized comfort models.

## Figures and Tables

**Figure 1 sensors-23-00576-f001:**
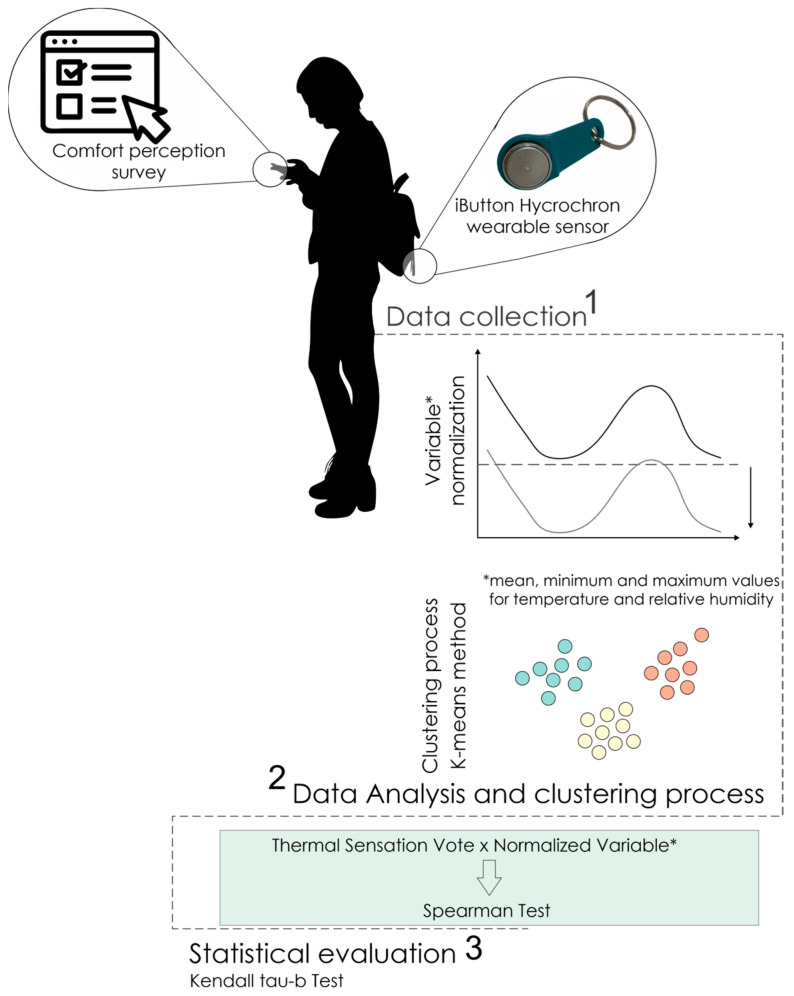
Research framework demonstrating steps 1 (data collection), 2 (data analysis and clustering process), and 3 (statistical calculation).

**Figure 2 sensors-23-00576-f002:**
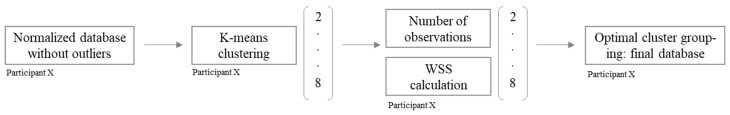
Cluster calculation procedure of the final database for each subject, from the normalized database to the final cluster grouping.

**Figure 3 sensors-23-00576-f003:**
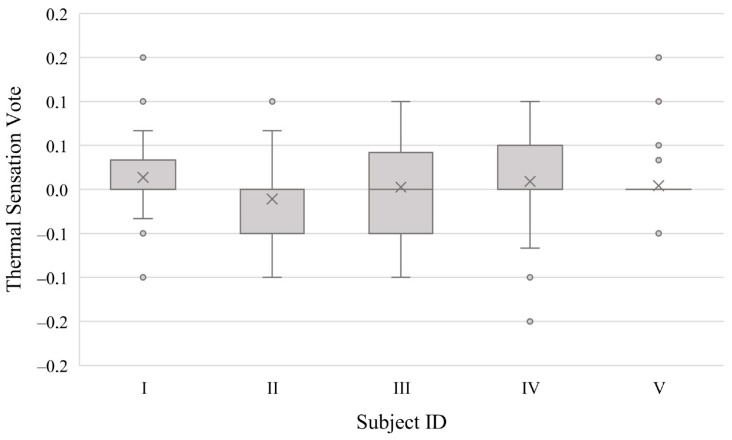
Thermal sensation of each subject, where the “x” represents the median, the points are the outliers, the box shows the interquartile range, and the bars demonstrate maximum and minimum calculated values within the sample.

**Figure 4 sensors-23-00576-f004:**
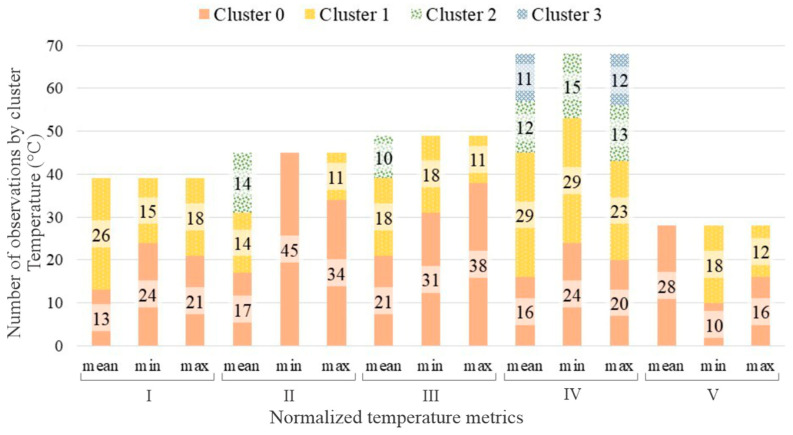
Stacked bars graph presenting the number of elements by cluster and temperature metric for each subject.

**Figure 5 sensors-23-00576-f005:**
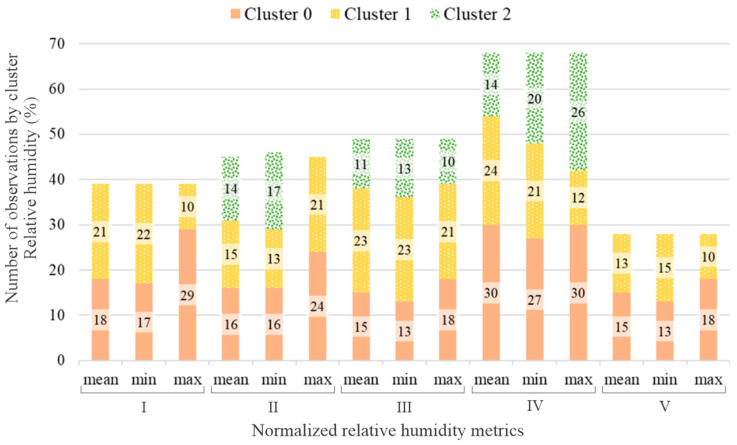
Stacked bars graph presenting the number of elements by cluster and relative humidity metric for each subject.

**Figure 6 sensors-23-00576-f006:**
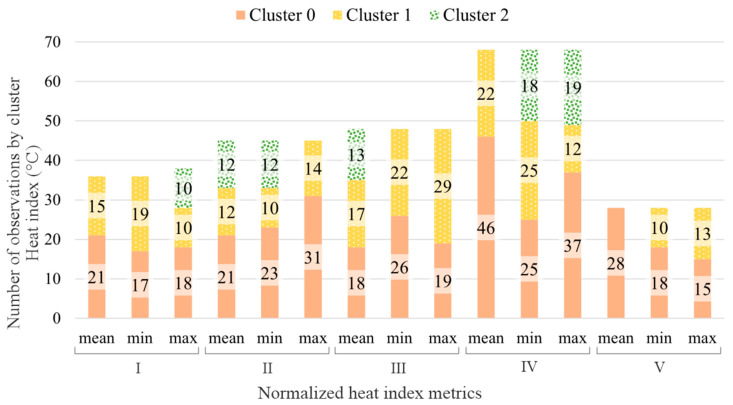
Stacked bars graph presenting the number of elements by cluster and heat index metric for each subject.

**Figure 7 sensors-23-00576-f007:**
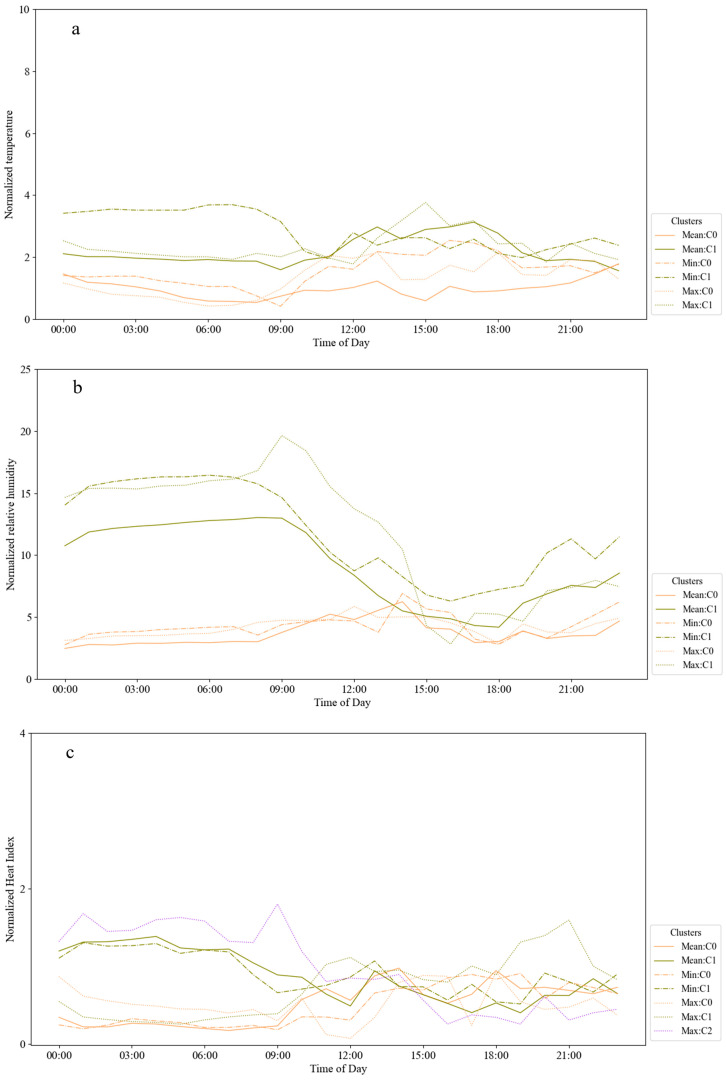
Daily temperature (**a**), relative humidity (**b**), and heat index (**c**) profiles for subject I. Each color line represents an index type, and the dash type shows the cluster.

**Figure 8 sensors-23-00576-f008:**
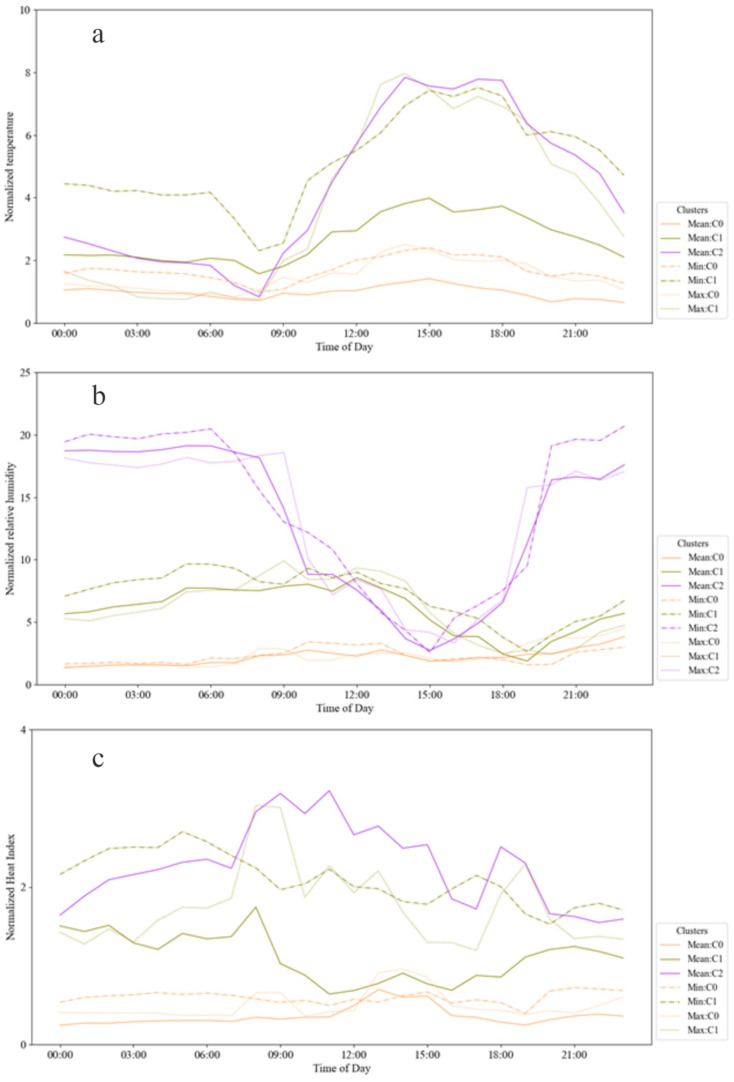
Daily temperature (**a**), relative humidity (**b**), and heat index (**c**) profiles for subject III. Each color line represents an index type, and the dash type shows the cluster.

**Table 1 sensors-23-00576-t001:** Adopted Thermal Sensation Vote scale.

−2	−1	0	+1	+2
Cold	Cool	Comfortable	Warm	Hot

**Table 2 sensors-23-00576-t002:** Start and end date of data collection for each subject.

Subject ID	Age	Metabolic Rate (W)	Start Date Hot Season	End Date Hot Season	Obs. in Hot Season (Days)	Start Date Cold Season	End Date Cold Season	Obs. in Cold Season (Days)	Received/Expected Answers (%)
I	34	506.13	18 August 2020	26 October 2020	45	04 March 2021	12 May 2021	30	48
II	26	450.88	11 August 2020	26 October 2020	52	01 March 2021	15 March 2021	15	66
III	31	603.00	12 August 2020	21 October 2020	45	02 March 2021	16 March 2021	15	82
IV	29	519.48	07 August 2020	21 October 2020	54	09 February 2021	09 March 2021	25	86
V	33	560.98	21 August 2020	26 October 2020	42	-	-	0	67
VI	-	-	05 August 2020	22 August 2020	18	24 February 2021	24 March 2021	29	21

**Table 3 sensors-23-00576-t003:** Within cluster average distance by metric for each subject.

Subject ID	Within Cluster Average Distance					
	Temperature			Relative Humidity		
	Mean	Minimum	Maximum	Mean	Minimum	Maximum
I	0.0274	0.0370	0.0290	0.0389	0.0373	0.0252
II	0.0130	0.0363	0.0130	0.0178	0.0171	0.0269
III	0.0121	0.0205	0.0112	0.0169	0.0183	0.0192
IV	0.0177	0.0192	0.0120	0.0169	0.0183	0.0192
V	0.0260	0.0169	0.0184	0.0326	0.0394	0.0292

**Table 4 sensors-23-00576-t004:** Kendall tau-b statistical calculations of temperature by subject and metric.

Temperature Values				
Metric	Cluster	Kendall Tau-b Correlation Coefficient	Kendall tau-b *p*-Value	Mean Value by Cluster
Subject I				
Mean	0	−0.031	0.893	1.162
	1	0.158	0.301	2.507
Minimum	0	−0.374	0.018	1.845
	1	0.000	1.000	3.124
Maximum	0	−0.196	0.247	1.453
	1	−0.035	0.856	2.820
Subject II				
Mean	0	−0.009	0.964	1.731
	1	0.035	0.866	1.725
	*2*	*−0.214*	*0.310*	*3.322*
Minimum	0	−0.048	0.667	2.709
Maximum	0	0.198	0.128	1.760
	1	0.081	0.742	3.485
Subject III				
Mean	*0*	*0.240*	*0.164*	*1.060*
	1	0.083	0.658	2.722
	2	−0.389	0.146	4.426
Minimum	0	−0.005	0.972	1.756
	1	−0.374	0.048	5.264
Maximum	0	0.115	0.357	1.539
	*1*	*−0.250*	*0.322*	*3.962*
Subject IV				
Mean	0	0.462	0.020	1.070
	1	−0.203	0.160	2.554
	*2*	*−0.263*	*0.259*	*4.982*
	3	0.038	0.874	5.609
Minimum	0	−0.040	0.795	2.426
	1	−0.145	0.307	5.263
	2	0.206	0.317	8.587
Maximum	*0*	*0.248*	*0.180*	*1.152*
	1	0.039	0.814	2.147
	2	0.113	0.576	3.357
	*3*	*0.230*	*0.321*	*4.725*
Subject V				
Mean	*0*	*0.302*	*0.046*	*1.878*
Minimum	0	−0.149	0.602	1.518
	1	0.107	0.573	4.062
Maximum	0	0.373	0.073	1.671
	1	0.040	0.869	2.727
*Moderate correlation*			Strong correlation	

**Table 5 sensors-23-00576-t005:** Kendall tau-b statistical calculations of relative humidity by subject and metric.

Relative Humidity Values				
Metric	Cluster	Kendall tau-b Correlation Coefficient	Kendall tau-b *p*-Value	Mean Value by Cluster
Subject I				
Mean	0	−0.104	0.594	3.949
	*1*	*0.236*	*0.163*	*10.259*
Minimum	0	0.064	0.745	4.718
	*1*	*0.327*	*0.051*	*13.129*
Maximum	0	−0.009	0.950	5.080
	1	0.511	0.054	12.329
Subject II				
Mean	0	0.135	0.503	6.424
	1	−0.010	0.959	10.633
	2	−0.079	0.720	14.012
Minimum	*0*	*0.221*	*0.290*	*8.509*
	*1*	*0.302*	*0.169*	*11.958*
	*2*	*−0.213*	*0.271*	*17.476*
Maximum	0	0.077	0.622	6.971
	1	0.006	0.974	13.429
Subject III				
Mean	0	−0.197	0.353	2.397
	1	−0.037	0.820	6.712
	2	0.059	0.809	13.637
Minimum	0	0.083	0.723	2.517
	1	−0.005	0.978	7.497
	2	0.072	0.746	15.068
Maximum	0	−0.173	0.364	2.916
	1	0.147	0.397	6.622
	2	−0.193	0.456	13.871
Subject IV				
Mean	0	−0.193	0.172	5.264
	1	0.068	0.663	11.721
	2	−0.188	0.371	15.108
Minimum	0	−0.116	0.429	6.696
	1	0.055	0.747	13.597
	2	−0.069	0.688	17.831
Maximum	0	−0.121	0.391	5.633
	1	0.509	0.026	10.007
	2	−0.014	0.927	12.924
Subject V				
Mean	0	0.000	1.000	8.996
	1	0.140	0.547	14.430
Minimum	0	0.105	0.652	14.355
	1	−0.138	0.521	21.433
Maximum	0	−0.082	0.675	6.730
	1	0.599	0.025	14.072
*Moderate correlation*			Strong correlation	

**Table 6 sensors-23-00576-t006:** Kendall tau-b statistical calculations of heat index by subject and metric.

Heat Index Values				
Metric	Cluster	Kendall tau-b Correlation Coefficient	Kendall tau-b *p*-Value	Mean Value by Cluster
Subject I				
Mean	0	−0.03475	0.839893	0.689227
	1	−0.42836	0.043394	1.115328
Minimum	0	0.016657	0.930244	0.615092
	1	−0.0684	0.715942	1.155734
Maximum	0	−0.32203	0.204976	0.785946
	1	0.196672	0.318203	0.880973
	2	−0.38103	0.154466	1.376008
Subject II				
Mean	*0*	*0.288315*	*0.097346*	*1.083491*
	1	0.197386	0.395355	2.215478
	*2*	*0.307255*	*0.193156*	*2.641658*
Minimum	*0*	*0.286592*	*0.078526*	*1.319569*
	1	0.052705	0.845009	2.356961
	2	0.407597	0.077839	3.379285
Maximum	*0*	*0.300322*	*0.031342*	*1.321422*
	1	0.318441	0.130206	2.830822
Subject III				
Mean	0	0.193589	0.309974	0.375035
	1	−0.36071	0.064451	1.330671
	2	−0.03269	0.890746	2.526225
Minimum	0	−0.0791	0.605212	0.653943
	*1*	*0.288806*	*0.091109*	*2.339653*
Maximum	0	0.329478	0.075592	0.519701
	1	−0.17795	0.22386	2.044502
Subject IV				
Mean	0	0.139446	0.210929	1.183001
	1	0.112792	0.486967	3.26676
Minimum	0	0.144039	0.349533	1.115322
	1	0.115728	0.454287	2.594446
	2	0.01452	0.937345	5.057425
Maximum	0	0.15183	0.224053	1.387462
	1	0.11134	0.642769	1.957179
	2	0.086838	0.618119	3.480653
Subject V				
Mean	*0*	*0.271538*	*0.072749*	*1.837843*
Minimum	0	0.106132	0.59567	1.803292
	*1*	*−0.25565*	*0.334428*	*4.172755*
Maximum	0	0.093906	0.672236	1.260548
	1	0.314945	0.160892	2.69809
*Moderate correlation*			Strong correlation	

## Data Availability

Not applicable.

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
