# Peer review of "Long-Term Thermal Comfort Monitoring via Wearable Sensing Techniques: Correlation between Environmental Metrics and Subjective Perception"

_sensors, 2023, doi:10.3390/s23020576_

Round 1
Reviewer 1 Report
I encourage the authors to expand introduction and discussion to include chemosensors for environmental monitoring (https://doi.org/10.1016/j.teac.2022.e00184) and compare it with other wearable sensors.
Additionally, authors need to elaborate the captions for each figure. Describe the content of figures in each caption.
For tables in addition to describing whether moderate or strong correlation, author should also provide information on whether these data are statistically significant or not.
Figure 1-could be redone by adding more details.
Author Response
- I encourage the authors to expand introduction and discussion to include chemosensors for environmental monitoring (https://doi.org/10.1016/j.teac.2022.e00184) and compare it with other wearable sensors.
R: Thank you for collaborating to expand the knowledge provide in this research. Chemosensors were included in the introduction in “Biosensors are another type of tools that could detect small toxic molecules, pollutants and life-threating agents present in the inhabited environments”. In the discussions, the sensor was approached providing a brief comparison with the wearable sensors used in this research and suggesting improvements for future works: “Other types of sensors can be adopted, measuring more parameters of the environment, such as biosensors”.
- Additionally, authors need to elaborate the captions for each figure. Describe the content of figures in each caption.
R: The authors thank the reviewer for this comment; this addition helps the readers to quickly identify the information contained in the Figures.
- For tables in addition to describing whether moderate or strong correlation, author should also provide information on whether these data are statistically significant or not.
R: Thank you for the indicating this missing information in the results section. The statistically significant values were indicated for tables 4, 5 and 6, respectively the Kendall tau-b results for temperature, relative humidity, and heat index values.
- Figure 1-could be redone by adding more details.
R: Following the recommendations, Figure 1 was modified adding information about the statistic test used, information about the clustering process method adopted and the three types of indices calculated.
Reviewer 2 Report
The manuscript entitled “Long-term thermal comfort monitoring via wearable sensing techniques: correlation between environmental metrics and subjective perception” is interesting. However, I have some comments or questions that need to be addressed before this manuscript could be accepted for publication.
1. The authors stated that “Table 1 presents the start and end date of data collection for each subject.” (Line 254). However, Table 1 shows the TSV Likert scale.
2. (Line 244): “The monitoring activity occurred in indoor and outdoor environments, for 24 hours, 7 days a week, for at least one month in different seasons of the year.” And (Line 254): “However, it is important to point out that some days between these dates were not registered due to inaccurate manipulation of the sensor by the participants.” If the participants manipulated the sensors inaccurately, how did the authors assess that during the dates on which the data were recorded, the sensors were not also inaccurately manipulated?
3. (Line 227): “The participants were instructed to leave the sensors visible to the outdoor environment and attached to accessories of daily use, such as key chains or backpacks.” The authors should include in the manuscript all the instructions provide to the participants. Is the instruction to leave the sensors visible to the outdoor environment sufficient?
4. Overall, the method section needs significant improvements. Authors should add and explain in the Section 2. Materials and Methods how they have ensured that the sensors were always in a correct position during data recording. Please report the locations of the physical measurements.
5. The main concern and limitation I find in this study is that the authors have analysed data recorded during periods when they do not know the exact location of the sensors. ANSI/ASHRAE Standard 55-2017 states that air temperature have to be measured at positions of 0.1m, 0.6m and 1.1m for seated participants, and 0.1m, 1.1m and 1.7m for standing participants. How did the authors know at what height the measurement was being taken? How did they know the participant's position at the time the air temperature was being measured?
6. In addition, why was not the mean radiant temperature measured? And the air velocity? Since the participants spent some periods outdoor, the mean radiant temperature shall be higher than ambient temperature and thus needs special considerations.
7. Regarding the questionnaire survey, were any questions about clothing levels included? The authors stated (line 363): “The daily participants' survey answers, for example, allowed the correlation between this subjective metric and environmental parameters, for the statistical test.” However, it is impossible to interpret the thermal comfort results without knowing the clothing levels.
8. Figures 4, 5 and 6 are confusing. The axis shows degrees Celsius and the scale exceeds the value of 60. Are these values correct?
9. Why were all the participants women?
10. “The mean heart rate, required for the calculation, was obtained from commercial smartwatches provided to the subject for the whole experimental campaign.” The authors should report the error and accuracy of the instruments used to measure the heart rate. Previous research claimed that wearable device such as smart-watches are arguably accurate due to the stability of the sensor (Maritsch, M., et al 2019).
Maritsch, M., Bérubé, C., Kraus, M., Lehmann, V., Züger, T., Feuerriegel, S., ... & Wortmann, F. (2019, September). Improving heart rate variability measurements from consumer smartwatches with machine learning. In Adjunct Proceedings of the 2019 ACM International Joint Conference on Pervasive and Ubiquitous Computing and Proceedings of the 2019 ACM International Symposium on Wearable Computers (pp. 934-938).
11. (Line 533): “The simple method allowed the inclusion of the wearable sensors in the particpants routine with minimal adjustments and almost any complaints on their side. However, some factors that could have influenced the results should be addressed, such as the possibility of inaccurate data collection, in the cases where the subjects left the sensors at home, registering temperatures with lower amplitude, even if the actual perceive temperature is different.” If factors such as the participants leaving their sensors at home have not been addressed in this study, the temperature and relative humidity values are not reliable and therefore, neither are the obtained results.
Author Response
Reviewer #2:
The manuscript entitled “Long-term thermal comfort monitoring via wearable sensing techniques: correlation between environmental metrics and subjective perception” is interesting. However, I have some comments or questions that need to be addressed before this manuscript could be accepted for publication.
- The authors stated that “Table 1 presents the start and end date of data collection for each subject.” (Line 254). However, Table 1 shows the TSV Likert scale.
R: Thank you for the detailed revision of the manuscript. The description was corrected, addressing the right table: “Table 1 presents the start and end date (…)”.
- (Line 244): “The monitoring activity occurred in indoor and outdoor environments, for 24 hours, 7 days a week, for at least one month in different seasons of the year.” And (Line 254): “However, it is important to point out that some days between these dates were not registered due to inaccurate manipulation of the sensor by the participants”. If the participants manipulated the sensors inaccurately, how did the authors assess that during the dates on which the data were recorded, the sensors were not also inaccurately manipulated?
R: Thank you for the observation that allowed us to clarify this point of the article’s method. The days that the subjects wrongly manipulated the sensors did not show temperature or relative humidity data registration and, because of that, the information of this period was not used. This data loss happened mainly due to data rollover (the collected information should have been downloaded first to avoid data loss). The term “inaccurate manipulation of the sensor” was changed to “data rollover issues”.
- (Line 227): “The participants were instructed to leave the sensors visible to the outdoor environment and attached to accessories of daily use, such as key chains or backpacks.” The authors should include in the manuscript all the instructions provide to the participants. Is the instruction to leave the sensors visible to the outdoor environment sufficient?
R: The participants were instructed to let the sensors visible to the outdoors and carry them in all daily situations, attached to backpacks, coats, keychains etc. Further explanation was also provided, describing that the device should be carry in this way in order to register reliable and realistic data. The sensor also should not be exposed to directly sunlight (to avoid overheating) or to immersed in water (considering that they are water-resistant, but not waterproof). All the instructions were included in the text: “The participants were instructed to leave the sensors visible to the outdoor environment and attached to accessories of daily use, such as key chains, or backpacks or even coats. It was also recommended that the sensors should not be exposed to direct sunlight, to avoid overheating, or immersed into waterbodies, considering that they are water-resistant, but not waterproof. Alarms were set to send reminders to the participants, according to the usual hour that they leave home”.
- Overall, the method section needs significant improvements. Authors should add and explain in the Section 2. Materials and Methods how they have ensured that the sensors were always in a correct position during data recording. Please report the locations of the physical measurements.
R: Thank you for addressing this topic, giving us the opportunity to better explain the used method. As explained in the method section and in the last question, the participants were instructed to constant monitor the position of the sensors; otherwise, considering that the sensors were worn during the whole day, do the continuous monitoring of the sensors’ position by the researchers was a laborious activity, if not impossible. According to the authors’ experience, here “correct position” for the sensors means that the thermo-hygrometer was in contact with the air and close to the subject, also not inside any bag, piece of clothing or even in a different room.
The location of the physical measurements, in this case meaning the city, was reported in page 7, in the paragraph: “The study was carried out in Perugia, Italy. According to the Köppen-Geiger criteria, Perugia has an overall mild climate, with no dry season and constantly moist, classified as a humid subtropical climate (Cfa) [21]”.
- The main concern and limitation I find in this study is that the authors have analysed data recorded during periods when they do not know the exact location of the sensors. ANSI/ASHRAE Standard 55-2017 states that air temperature have to be measured at positions of 0.1m, 0.6m and 1.1m for seated participants, and 0.1m, 1.1m and 1.7m for standing participants. How did the authors know at what height the measurement was being taken? How did they know the participant's position at the time the air temperature was being measured?
R: Taking the comments made in the response for question 4, and considering that it is a long-term monitoring, and that the participants were monitored during their daily activities, the complexity to measure the sensors’ height for monitoring temperature and RH is not justified and yet very difficult to control. Other similar works, such as Liu et al. [1], presented the same type of device and measurement approach, showing reliable results for temperature and relative humidity measurements. Other than that, the ANSI/ASHRAE Standard 55-2017 temperature measurement fits better for controlled test, where the subject is not in movement or in a test room for example. Here, we were not measuring the room environmental quality or the PMV/PPD indexes, but the effect of the external environment in the person. Considering the specific aspect of these research, the constantly adjustment of the sensor height would not possible.
- In addition, why was not the mean radiant temperature measured? And the air velocity? Since the participants spent some periods outdoor, the mean radiant temperature shall be higher than ambient temperature and thus needs special considerations.
R: In this research, a wearable sensor was employed seeking to impact as little as possible in the participants’ daily activities, prioritizing some characteristics such as long term resistance, convenience, and portability. Considering the commercial wearable sensors to monitor mean radiant temperature and air velocity [2], it was preferable to not imposed their use, for the sue to the issues in the measurements and that they would probably not be constant. Also, some issues could be faced for the measurement of mean radiant temperature outdoors, requiring solar radiance, long and short waves length data, conversion calculations and a careful positioning of the sensing system. However, new research is being developed to create new wearable devices with this purpose, that could be applied in future works, mainly for controlled tests [3-5].
- Regarding the questionnaire survey, were any questions about clothing levels included? The authors stated (line 363): “The daily participants' survey answers, for example, allowed the correlation between this subjective metric and environmental parameters, for the statistical test.” However, it is impossible to interpret the thermal comfort results without knowing the clothing levels.
R: Clothing level questions were not addressed considering that participants had similar routines and behaviour towards this topic, making the influence of this question minimal to comfort perception. Also, considering that the survey should be answered three times a day, a simple and quick questionary was preferred to keep participants engaged. Another observation that should be made is that the collected data was not used in PMV/PPD calculations, which require the clothing level.
- Figures 4, 5 and 6 are confusing. The axis shows degrees Celsius and the scale exceeds the value of 60. Are these values correct?
R: Thanks for the question, helping to make clearer to readers this topic. The type of graph presented in Figures 4, 5 and 6 is the “stacked bars”, were the columns or bars are stacked on top of each other. The value of each variable is the difference between the initial and final values reported in the y-axis. A clarification on the figure caption was added to explain this point and avoid doubts about the graph typology.
- Why were all the participants women?
R: The participants were all women seeking to present a more homogeneous sample, since the number of people participating was relatively low. As presented in the conclusion section, as a future development, it is planned to increase the number of participants, varying gender, age, activity level and routine.
- “The mean heart rate, required for the calculation, was obtained from commercial smartwatches provided to the subject for the whole experimental campaign.” The authors should report the error and accuracy of the instruments used to measure the heart rate. Previous research claimed that wearable device such as smart-watches are arguably accurate due to the stability of the sensor (Maritsch, M., et al 2019).
R: Thank you for the observation. The information concerning the smartwatch type and its accuracy was included in the manuscript: “The mean heart rate, required for the calculation, was obtained from the Samsung Galaxy smartwatch provided to the subject for the whole experimental campaign. According to previous research, the device has an accuracy around +/- 4bpm [6,7]”. The model employed in the research was the first Samsung Galaxy watch released, but the accuracy reported was the same from the second model (considering the modifications made from one model to another).
- (Line 533): “The simple method allowed the inclusion of the wearable sensors in the particpants routine with minimal adjustments and almost any complaints on their side. However, some factors that could have influenced the results should be addressed, such as the possibility of inaccurate data collection, in the cases where the subjects left the sensors at home, registering temperatures with lower amplitude, even if the actual perceive temperature is different.” If factors such as the participants leaving their sensors at home have not been addressed in this study, the temperature and relative humidity values are not reliable and therefore, neither are the obtained results.
R: The authors would like to thank the reviewer for this observation, allowing us to address the matter in a clearer way. The mentioned paragraph has some information missing: the authors warn the participants that, if they accidentally leave their sensors at home or simply do not have it with them, they shouldn’t answer the questionary. Only the days with answers were considered to perform the analysis in the research. The following phrase was added: “In this research the participants were instructed to answer the questionary just having it with them, in order to avoid such problems”. Nevertheless, the authors could not directly verify whether subjects were fair in answering or wearing the monitoring device. This could be a limitation but also a suggestion for properly design the recruitment procedure, as further highlighted in the conclusion section: “Given the human-centric approach employed in this campaign, strategies to keep the participants engaged in it, i.e., wearing the sensors and answering the survey, should be implemented, guaranteeing serious and active participation of the subjects.”.
We would like to thank the editor and the reviewers for all contribution and important observations provided for this research. Along with this letter, we also forward to you the final text, with the corrections tracked.
References
[1] S. Liu, S. Schiavon, H.P. Das, M. Jin, C.J. Spanos, Personal thermal comfort models with wearable sensors, Build Environ. 162 (2019) 106281. https://doi.org/10.1016/j.buildenv.2019.106281.
[2] https://www.az-instrument.com.tw/en/product-616308/Relative-Humidity-Anemometer-8902-AZ.html
[3] Pigliautile, I.; Pisello, A.L. A new wearable monitoring system for investigating pedestrians’ environmental conditions: Development of the experimental tool and start-up findings. Sci. Total Environ. 2018, 630, 690–706.
[4] Pioppi, B.; Pigliautile, I.; Pisello, A.L. Data collected by coupling fix and wearable sensors for addressing urban microclimate variability in an historical Italian city. Data Brief 2020, 29, 105322.
[5] Nakayoshi, M.; Kanda, M.; Shi, R.; De Dear, R. Outdoor thermal physiology along human pathways: A study using a wearable measurement system. Int. J. Biometeorol. 2015, 59, 503–515.
[6] M. Nissen, S. Slim, K. Jäger, M. Flaucher, H. Huebner, N. Danzberger, P.A. Fasching, M.W. Beckmann, S. Gradl, B.M. Eskofier, Heart Rate Measurement Accuracy of Fitbit Charge 4 and Samsung Galaxy Watch Active2: Device Evaluation Study, JMIR Form Res. 6 (2022) e33635. https://doi.org/10.2196/33635.
[7] J. Hwang, J. Kim, K.-J. Choi, M.S. Cho, G.-B. Nam, Y.-H. Kim, Assessing Accuracy of Wrist-Worn Wearable Devices in Measurement of Paroxysmal Supraventricular Tachycardia Heart Rate, Korean Circ J. 49 (2019) 437. https://doi.org/10.4070/kcj.2018.0323.
Round 2
Reviewer 1 Report
Authors have addressed most of the comments.
Reviewer 2 Report
The authors have properly addressed my previous comments.